# Estimated impact of the 2020 economic downturn on under-5 mortality for 129 countries

Marcelo Cardona[1,6]ʘ*, Joseph Millward[2,5]ʘ, Alison Gemmill[3]ʘ, Katelyn Jison Yoo[4,5]‡, David M. Bishai[5]‡

1 ROCKWOOL Foundation, Copenhagen C, Denmark, 2 Johns Hopkins University Center for Communications Programs, Baltimore, Maryland, 3 Department of Population, Family and Reproductive Health, Johns Hopkins Bloomberg School of Public Health, Baltimore, Maryland, 4 Health, Nutrition, and Population, World Bank, Washington, DC, United States of America, 5 Johns Hopkins Bloomberg School of Public Health, Baltimore, Maryland, 6 Institute for Advanced Development Studies (INESAD), La Paz, Bolivia

ʘ These authors contributed equally to this work.
‡ KJY and DMB also contributed equally to this work.
* mcc@rff.dk

**Data Availability Statement:** The data underlying the results presented in the study are available from United Nation's World Population Prospects 2019 Revision (https://population.un.org/wpp/#:~:text=The%202019%20Revision%20of%25

## Abstract

In low- and middle-income countries (LMICs), economic downturns can lead to increased child mortality by affecting dietary, environmental, and care-seeking factors. This study estimates the potential loss of life in children under five years old attributable to economic downturns in 2020. We used a multi-level, mixed effects model to estimate the relationship between gross domestic product (GDP) per capita and under-5 mortality rates (U5MRs) specific to each of 129 LMICs. Public data were retrieved from the World Bank World Development Indicators database and the United Nations World Populations Prospects estimates for the years 1990-2020. Country-specific regression coefficients on the relationship between child mortality and GDP were used to estimate the impact on U5MR of reductions in GDP per capita of 5%, 10%, and 15%. A 5% reduction in GDP per capita in 2020 was estimated to cause an additional 282,996 deaths in children under 5 in 2020. At 10% and 15%, recessions led to higher losses of under-5 lives, increasing to 585,802 and 911,026 additional deaths, respectively. Nearly half of all the potential under-5 lives lost in LMICs were estimated to occur in Sub-Saharan Africa. Because most of these deaths will likely be due to nutrition and environmental factors amenable to intervention, countries should ensure continued investments in food supplementation, growth monitoring, and comprehensive primary health care to mitigate potential burdens.

## 1 Introduction

Economic downturns have occurred in almost all countries as a result of COVID-19. We know from prior research that these economic downturns have a disproportionate effect on child health and mortality in low- and middle-income countries (but not high-income countries) and that these effects are likely independent of whether or not children acquire COVID-19 disease. [1–4].

20World%20Population%20Prospects%20is,and
%20Social%20Affairs%20of%20the%20United%
20Nations%20Secretariat) and from the World
Bank World Development Indicators (https://
databank.worldbank.org/reports.aspx?source=
world-development-indicators).

**Funding:** This work was funded by the
ROCKWOOL Foundation in the form of a grant to
MC [grant no. 1227]. The funders had no role in
study design, data collection, and analysis,
decision to publish, or preparation of the
manuscript.

**Competing interests:** The authors have declared
that no competing interests exist.

GDP per capita is a remarkably robust social determinant of health in multivariate analyses.
[1] The empirical relationship between mortality and national income was first noted by Samuel Preston, who found that there is a positive log relationship between a country's gross domestic product (GDP) per capita and life expectancy (i.e., the "Preston curve"). [3] Many subsequent studies have confirmed the relationship and have shown that there is a more significant effect of recessions occurring in countries with lower levels of GDP and higher income inequality. [2, 3, 5, 6] Recent studies have confirmed the adverse effect of recessions on under-5 mortality, showing that the impact in low- and middle-income countries (LMICs) is three times larger than in countries with better economic indicators. [7–10] While there are variable effects of recessions on mortality noted across varying socio-economic contexts, there is limited research that examines the relationship between child mortality and economic downturns caused by a pandemic in LMICs.

The mechanisms relating social health determinants like GDP per capita to child mortality in low- and middle-income countries are presumed to act through the combined effects of environmental contamination, nutrient deficiency, maternal factors, injury, and personal illness control. [4] For example, reductions in household income can unleash dual effects of environmental contamination and nutrient deficiency as households cope with poverty by moving to poorer housing with less sanitation and more crowding, as well as shifting diets away from costly sources of protein, and micronutrients. [11] A spiral of successive gastrointestinal, skin, and respiratory infections can further deplete nutritional reserves. Parents securing supplemental income during economic hardship can subject children to less parental supervision heightening the risk of injury. [12, 13] Both the demand for and the supply of essential childhood health services including immunizations, micronutrients, and primary care can falter during a severe economic downturn. Effects of economic downturns on health are considered indirect as they manifest through direct determinants of health such as access to and quality of food, water, and housing, and additionally limit the resources available for expenditure on healthcare. [4].

Knowing how the global 2020 economic downturn led to worsening child health and mortality can help inform policymakers, clinicians, and advocates about the magnitude of such effects and can improve strategies to reduce disease burden, especially in the face of a prolonged pandemic. To our knowledge, however, no study has estimated the indirect health effects of the 2020 economic downturn, even though past economic downturns have been shown to lead to health declines, especially among children. For example, studies from Ebola outbreaks in Africa and SARS in East Asia have highlighted the importance of national and international organizations in combating the indirect economic effects of disease on the most economically disadvantaged communities. [14, 15] Further, a systematic review of the social and economic burden of seasonal influenza in LMICs found that influenza's indirect costs, namely productivity loss, were significantly higher in LMICs than high-income countries. [16].

Prior studies of the indirect death toll due to an epidemic-related downturn may not be relevant to 2020 because prior epidemics did not spur an economic slowdown of the same magnitude as those experienced in 2020. Estimates indicate that the world economy was expected to shrink more than 5% in 2020 alone. [17–20] The economic downturns of 2020 have also been projected to reverse a sustained trend of decline in global poverty, with an estimated 42–66 million additional children falling into extreme poverty on top of the estimated 386 million children in extreme poverty in 2019. [21, 22] Additional estimates suggest that the economic effects of the COVID-19 pandemic could reverse the past 2 to 3 years of progress in infant mortality. [22].

This study assesses the indirect economic effects of the COVID-19 pandemic by estimating the impact of different economic downturn scenarios on under-5 mortality in low, lower-middle, and upper-middle-income countries. While there is some uncertainty about the final magnitude of the economic downturns of 2020, our model projects excess under-5 mortality by country that are relevant for reductions in GDP per capita in 2020, as compared to 2019 baseline values, as small as 5% and as large as 15% to be used as reference points. Our approach draws from the empirical relationship between mortality and national income that was first noted by Preston and has been widely documented. [3].

## 2 Methods

### 2.1 Overview and data sources

The methodology is presented in three sections. In section one, we present the methods used to re-estimate and update Preston curves specific to each LMIC using data from 1990 to 2020; whereas the original Preston uses life expectancy as a summary measure of health, here we use the under-5 mortality rate. [23] Specifically, the Preston curves we generate provide multivariate adjusted estimates of the slope parameter relating GDP and Under-5 mortality individualized to each country's most recent data along with 95% confidence intervals. In the second section, we apply each country's GDP-Under-5 mortality slope parameter to estimate the one-year mortality impact of a 5%, 10%, and 15% reduction in GDP. Finally, using Monte Carlo methods, we obtain uncertainty ranges around these excess mortality projections.

Because economic downturns in high income countries are known to have less widespread effects on child mortality, we only included 129 countries that were classified as low-, lower-middle-, or upper-middle income. Our study is based on the 2020 World Bank income classification requiring countries to have a gross national income (GNI) per capita below 12,375 US$. [24] Annual estimates of under-5 mortality for each country were obtained from the United Nation's World Population Prospects 2019 Revision. Data on GDP per capita (constant 2010 US$) were obtained from the World Bank World Development Indicators database—all the data for this study were retrieved in September 2020. Covariates include country-year-specific characteristics and health-specific services obtained from the World Bank World Development Indicators database. We imputed missing values in GDP per capita using a five-year moving average and in some covariates using multivariate normal regression. (See S1 Appendix for additional description of our imputation approach.) All estimated effects of economic downturns on under-5 mortality were calculated within a one-year time horizon, meaning that the increased mortality rates presented are representative of different downturn scenarios reflecting reductions in GDP per capita in 2020, as compared to 2019 baseline GDP per capita.

### 2.2 Multilevel mixed effects multivariable regression analysis

Regression analysis was used to estimate the Preston curve relationship between national income and under-5 mortality. First, we regressed the U5MR on GDP per capita and a set of socio-economic covariates. A model-based approach using an iterative process was used to fill in missing values in the set of covariates. For more details please see S2 Appendix. To estimate country-specific effects of a downturn, we applied a multilevel mixed-effect linear regression to the relationship between GDP per capita and U5MR for each country. [25–28] The rationale for using a multilevel mixed-effect model is because it allows us to control for heterogeneity across countries and to included fixed effects for a country's region and income level. (Sensitivity analyses showed that results were not sensitive to inclusion of fixed effects.) A log-log-linear mixed-effect model was estimated to ease the retransformation of impacts from a log-scale to natural units. This specification has been used to have a linear relationship between

U5MR and GDP per capita, and to represent the elasticity of U5MR with respect to GDP per-capita. Estimates were bracketed at 5%, 10%, and 15% reductions in country GDP per capita. Our baseline model to estimate the relationship between GDP per capita and U5MR had the following form:

$$\text{logU5mr}_{j,t} = \alpha_j + \beta_{1,j} logGDP_{j,t} + \epsilon_{j,t} \tag{1}$$

Where $\beta_{1,j}$ captures a country-specific relationship between GDP and under-5 mortality for years t = 1990–2020 in country $j$. The intercept $\alpha_j$ represents both the fixed and random intercept for country j. The residual represented by $\epsilon_{j,t}$ captures the error term for country j at time t. Because Eq 1 might omit other factors that are closely related to changes in the under-5 mortality rate, we extend our model presented in Eq 1 to include other country-specific factors that could affect the relationship between GDP and U5MR, as shown in Eq 2 below:

$$\text{logU5mr}_{j,t} = \alpha_j + \beta_{1,j} logGDP_{j,t} + \beta_2 Z_{j,t} + \beta_3 H_{j,t} + \epsilon_{j,t} \tag{2}$$

Where $Z_{j,t}$ represents a vector of country-year-specific characteristics. These control variables were as follows: electric power consumption per capita, the proportion of seats held by women in national parliaments, and total fertility rate for country. The last vector of controls $H_{j,t}$ captures health-specific services for each country-year and includes: the number of physicians per thousand inhabitants and the share of children (between 12 and 23 months) who had been immunized with a diphtheria pertussis and tetanus vaccine (DPT). By measuring GDP effects on mortality net of immunization coverage, our final model (Eq 2) isolates the GDP mortality effect primarily through effects on wasting, non-vaccine-related diseases, as well as parental caregiving and injury.

## 2.3 Lives lost estimation

Country-specific estimates of $\beta_{1,j}$ were then applied to GDP per capita data to predict an estimated mortality impact under the four different scenarios—no reduction in GDP per capita (scenario 1), 5% reduction (scenario 2), a 10% reduction (scenario 3), and 15% (scenario 4). These estimates were then compared to scenario 1, which represents baseline under-5 mortality. By separately subtracting estimated deaths from scenarios 2 to 4 from those in scenario 1 (i.e. no GDP per capita reduction) we are able to provide estimates of additional lives lost that are attributable to each level of economic downturn.

## 2.4 Estimates of uncertainty

We carried out sensitivity analyses to examine the uncertainty range of our estimates by sampling from a normal distribution parameterized with a mean and standard error of $\beta_{1,j}$ at the country level. In doing so, we performed a Monte Carlo experiment using 500 iterations to draw each country's GDP-U5MR impact parameter from normal distributions based on estimates of the coefficient and standard error estimated from Eq 2. For the simulation, the estimated log(U5MR) from each scenario was retransformed to a mortality rate and then multiplied by the population of children under-5 to produce an estimated total number of deaths under each scenario. For a graphical description of the results from the simulations for each scenario, please see S7 Appendix. The means and standard deviations of the incremental death projections emerging from each sample of 500 iterations is shown in Table 3. Because the Monte Carlo results emerge from 500 iterations, they differ slightly from the single iteration estimates.

# 3 Results

Between 1990 and 2019, there has been a sustained trend of decline in global poverty and infant mortality in LMICs. However, as hypothesized above, COVID-19 related economic downturns of 2020 are likely to reverse these positive trends. Table 1 presents select summary statistics for variables used in the analysis for the years 2010, 2015, and 2019. (S1 Appendix presents annual statistics for the entire study period, 1990–2020. Values for 2020 are based on projections from various sources that do not take into account the 2020 pandemic).

## 3.1 Under-5 mortality

The results from fitting models of U5MR and GDP for each country are shown in S3–S5 Appendices. Our baseline projection is a benchmark where there is no reduction in GDP per

**Table 1. Descriptive statistics of main variables in the sample of 129 LMIC countries (values prior to imputation).**

| | Year | | |
|---|---|---|---|
| | **2010** | **2015** | **2019** |
| **Under-five mortality rate (deaths under age 5 per 1,000 live births)** | | | |
| Mean | 66.69 | 43.52 | 38.37 |
| Standard Deviation | 42.26 | 32.34 | 28.81 |
| Share of missing observations | 0 | 0 | 0 |
| **GDP per capita constant 2010$** | | | |
| Mean | 2,973 | 3,738 | 3,996 |
| Standard Deviation | 3,306 | 3,163 | 3,273 |
| Share of missing observations | 2.33 | 4.65 | 10.85 |
| **Physicians (per 1,000 people)** | | | |
| Mean | 0.61 | 1.19 | 1.17 |
| Standard Deviation | 0.9 | 1.43 | 1.37 |
| Share of missing observations | 17.05 | 53.49 | 100 |
| **Electric power consumption (kWh per capita)** | | | |
| Mean | 955 | 1,384 | 1,467 |
| Standard Deviation | 1,279 | 1,366 | 1,369 |
| Share of missing observations | 32.56 | 100 | 100 |
| **Proportion of seats held by women in national parliaments (%)** | | | |
| Mean | 14.8 | 19.76 | 22.65 |
| Standard Deviation | 10.7 | 12.16 | 12.33 |
| Share of missing observations | 3.88 | 2.33 | 0.78 |
| **Total fertility (live births per woman)** | | | |
| Mean | 4.1 | 3.28 | 3.11 |
| Standard Deviation | 1.51 | 1.35 | 1.24 |
| Share of missing observations | 0 | 0 | 0 |
| **Immunization, DPT (% of children ages 12–23 months)*** | | | |
| Mean | 81.69 | 84.74 | 88.24 |
| Standard Deviation | 15.6 | 16.37 | 18.41 |
| Share of missing observations | 0.78 | 0 | 100 |

Source: Authors' elaboration

Table Notes: For a detailed description for every year, see S1 Appendix.

* Child immunization, DPT, measures the percentage of children ages 12–23 months who received DPT vaccinations before 12 months or at any time before the survey. A child is considered adequately immunized against diphtheria, pertussis (or whooping cough), and tetanus (DPT) after receiving three doses of vaccine. [22]

capita (i.e., Scenario 1), and in this case the expected total number of annual under-5 lives lost in LMICs would be around 19.2 million. Under a conservative scenario (5% reduction on GDP per capita; Scenario 2), the total number of under-5 deaths increases to 19.5 million, or an additional 282,996 number of deaths (95% CI: 279,779–286,400). The results for each scenario at the country level suggest that for the scenarios of 10% and 15% GDP reductions, there is an estimated under-5 loss of life of 19.8 and 20.2 million, which corresponds to an additional 585,802 (95% CI: 579,184–592,799) and 911,026 (95% CI: 900,804–921,825) lives lost, respectively. Moreover, we estimate that 49% of the total under-5 lives lost would occur in Sub-Saharan Africa, a pattern that is observed across the four scenarios, where the total number of lives lost in this region increased up to over 470,000 between a no downturn scenario and a 15% reduction in GDP per capita.

The estimated number of deaths is the largest in countries with a higher population. Consequently, Table 2 presents results for the ten countries with the highest additional under-5 lives lost in 2020 under the four different scenarios. Results suggest that India will be the country with the highest number of under-5 lives lost, followed by Nigeria and the Democratic Republic of the Congo. Furthermore, in countries like Burundi, Niger and the Democratic Republic of the Congo, a 5% reduction of GDP per capita represents a loss of 9.7, 9.6 and 8.8 percent of the total under-5 population, respectively. The estimates for the top 10 countries with the highest under-5 mortality rates are presented in S6 Appendix.

Fig 1 presents the number of total additional deaths from a 15% reduction in GDP per capita (e.g. Scenario 4), according to income group classification. Results show that a 15% reduction in GDP per capita will have a substantial increase in the under-5 mortality rate in LMICs, with larger estimated impacts in lower-middle income countries, where under-5 mortality rates tend to be higher.

## 3.2 Sensitivity analysis and robustness

Table 3 presents the results from a Monte Carlo experiment on the estimated logarithm of U5MR for each country in every scenario. Moreover, S7 Appendix presents a graphical description of the results from the simulations for each scenario. Thus, we observe that our estimations remain within the 95 per cent confidence interval across all scenarios, thereby validating the robustness of our approach.

## 4 Discussion

We estimate that the economic downturns of 2020 significantly increased loss of life among children younger than five years old in LMICs. Many of the countries in this analysis have relatively young populations with tenuous access to stable housing, clean water, food, and primary care. The health of these children is highly susceptible to reductions in the economic wellbeing of their families. Children in these lower income countries are also subject to a high rate of exposure to other infectious diseases, besides COVID-19, which makes them more susceptible when the economy reduces their access to nutrition, housing, water, sanitation, and parental care.4 Disruptions to primary health care service supply and demand will compound these threats, and thus may be a likely driver of increased mortality in these settings. Efforts to shore up the delivery of pediatric primary health care services during an economic downturn can mitigate the mortality impact of a downturn.

Our estimates match the lower range of other estimates of the indirect effects of the COVID-19 pandemic on child mortality which have primarily focused on excess mortality attributed to disruptions in delivery of key health services affecting children and mothers. Admittedly, this may primarily be driven by exclusion of delayed mortality effects after one

**Table 2. Estimated under-five lives lost from 2020 downturns scaled from 5% to 15%.**

| Country | Under 5 deaths | Lower bound (95% CI) | Upper bound (95% CI) | Under 5 deaths 5% reduction on GDP | Additional deaths 5% | Lower bound (95% CI) | Upper bound (95% CI) | Under 5 deaths 10% reduction on GDP | Additional deaths 10% | Lower bound (95% CI) | Upper bound (95% CI) | Under 5 deaths 15% reduction on GDP | Additional deaths 15% | Lower bound (95% CI) | Upper bound (95% CI) |
|---|---|---|---|---|---|---|---|---|---|---|---|---|---|---|---|
| India | 2,929,298 | 986,082 | 8,701,895 | 2,972,361 | 43,063 | 1,004,659 | 8,793,951 | 3,018,437 | 89,139 | 1,024,604 | 8,892,182 | 3,067,926 | 138,628 | 1,046,101 | 8,997,378 |
| Nigeria | 1,503,219 | 497,646 | 4,540,714 | 1,525,317 | 22,098 | 507,077 | 4,588,238 | 1,548,962 | 45,743 | 517,205 | 4,638,937 | 1,574,358 | 71,139 | 528,124 | 4,693,221 |
| Democratic Republic of the Congo | 1,388,004 | 524,706 | 3,671,682 | 1,408,409 | 20,405 | 534,338 | 3,712,285 | 1,430,241 | 42,237 | 544,670 | 3,755,652 | 1,453,691 | 65,687 | 555,796 | 3,802,143 |
| China | 1,235,908 | 372,924 | 4,095,918 | 1,254,076 | 18,169 | 380,064 | 4,138,001 | 1,273,517 | 37,609 | 387,734 | 4,182,878 | 1,294,396 | 58,489 | 396,005 | 4,230,905 |
| Pakistan | 1,054,683 | 371,239 | 2,996,334 | 1,070,187 | 15,505 | 378,185 | 3,028,413 | 1,086,777 | 32,094 | 385,641 | 3,062,651 | 1,104,595 | 49,912 | 393,676 | 3,099,329 |
| Ethiopia | 992,985 | 369,329 | 2,669,755 | 1,007,582 | 14,598 | 376,148 | 2,698,999 | 1,023,202 | 30,217 | 383,463 | 2,730,228 | 1,039,977 | 46,993 | 391,343 | 2,763,698 |
| United Republic of Tanzania | 523,317 | 187,417 | 1,461,240 | 531,010 | 7,693 | 190,917 | 1,476,931 | 539,241 | 15,925 | 194,674 | 1,493,681 | 548,083 | 24,766 | 198,723 | 1,511,625 |
| Indonesia | 461,840 | 147,090 | 1,450,106 | 468,629 | 6,789 | 149,891 | 1,465,156 | 475,893 | 14,054 | 152,899 | 1,481,208 | 483,696 | 21,856 | 156,142 | 1,498,392 |
| Niger | 461,338 | 171,118 | 1,243,775 | 468,120 | 6,782 | 174,273 | 1,257,434 | 475,377 | 14,039 | 177,657 | 1,272,020 | 483,171 | 21,833 | 181,302 | 1,287,654 |
| Bangladesh | 435,117 | 153,268 | 1,235,269 | 441,513 | 6,396 | 156,135 | 1,248,500 | 448,358 | 13,241 | 159,212 | 1,262,622 | 455,709 | 20,592 | 162,528 | 1,277,750 |

Source: Authors' elaboration

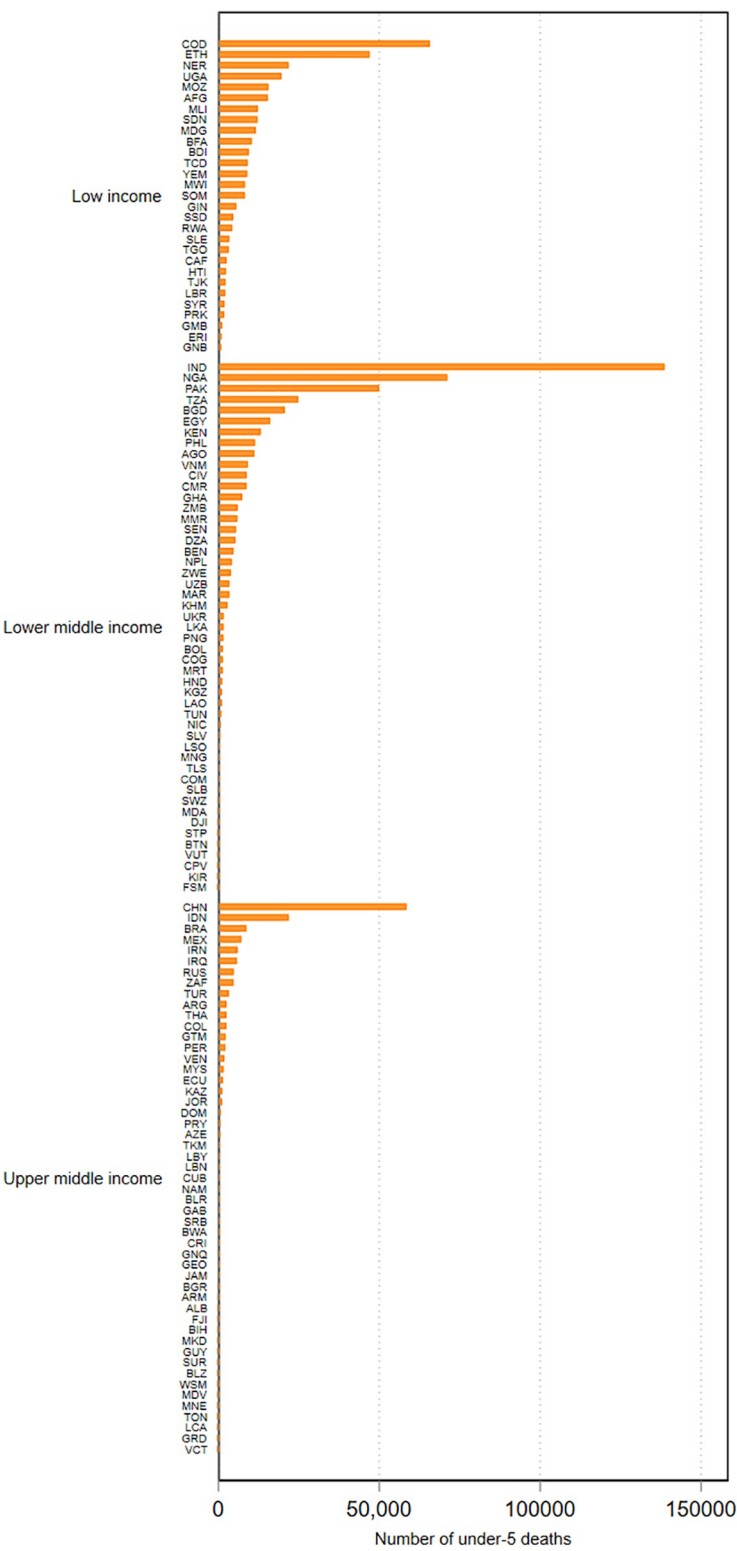

**Fig 1. Changes in under-five mortality from a 15% downturn, by country and income group.** Source: Authors' elaboration.

**Table 3. Uncertainty analysis from a Monte Carlo experiment using estimates of Model 2.**

| Scenario | Model 1 | Model 2 | Monte Carlo Version of Model 2 Mean (SD) | | Monte Carlo Version of Model 2 95% CI | |
|---|---|---|---|---|---|---|
| | | | | | Lower bound | Upper bound |
| 5% Recession | 402,847 | 282,996 | 283,090 | -1,689 | 279,779 | 286,400 |
| 10% Recession | 837,922 | 585,802 | 585,991 | -3,473 | 579,184 | 592,799 |
| 15% Recession | 1,309,822 | 911,026 | 911,314 | -5,362 | 900,804 | 921,825 |

Source: Authors' elaboration

year, economically mediated deaths in adults, and non-fatal effects on health, social development, and cognition that are known to follow famines and adverse childhood experiences. However, estimates of just the mortality effects of the 2020 downturns can help alert policymakers of the need to plan additional efforts to mitigate the economic threats faced by vulnerable groups. Reductions in service delivery could range between 10–52% and the prevalence of wasting could increase by 10–50%. [29] The estimated death toll due to health service reductions was estimated to range from 253,500 to 1,157,000 additional child deaths over a 6 month period with 60% of these deaths, linked to reduced coverage of childbirth services and 18–23% of deaths tied to wasting.26 Another study which focused on malaria service delivery disruption found that 25%–75% reductions in coverage of preventative and curative supplies and care may result in anywhere from 23,600 to 382,100 additional deaths in the most and least conservative scenarios, respectively. [30] In comparison, our analysis finds that 5%–15% reductions in GDP are estimated to lead to additional loss of life in children under five between 282,996 to 911,026. Our estimates are focused on those due to the reduction in GDP and do not include any direct effects of COVID-19 on children. Because our model controls for DPT vaccine delivery (i.e., our model assumes that DPT vaccine delivery is fixed) it underestimates the potential impact of economic downturn through these secondary effects on services. We find that the estimated additional lives lost from 5% and 15% downturns would equate to 1.5% and 4.7% increases above baseline, respectively.

The uncertainty surrounding the actual intensity and duration of COVID-19-induced economic effects is a significant limitation of this study. The study aimed to control for uncertainty by offering a bracketed range of likely economic downturn magnitudes from 5% to 15%, which allows countries to situate their own estimated economic downturn rates within this range to customize results.

Further limitations exist in the data that were used in this study. For example, many observations from the United Nations Inter-agency Group for Child Mortality Estimation and World Bank World Development Indicators required imputation up to 2020. Measurement of U5MR in many LMICs cannot be based on vital registration systems and must be based on demographic models of survey data produced by the United Nations. Authors also recognize alternative data sources for child mortality such as those available from the University of Washington Institute for Health Metrics and Evaluation, and acknowledge that both datasets are widely used in global health research. In addition, the study only focuses on the lives lost to children under-5 and does not examine other short- and long-term health-related impacts due to COVID-19 related economic downturns. Further research should focus on the non-fatal health effects of the 2020 economic downturns on health, cognition, development, and school attainment.

The empirical evidence correlating health and wealth initially outlined by Samuel Preston, and later expanded by authors such as Angus Deaton, highlighted that mortality in children

under-5 is one of the most significantly affected health outcomes from changes in GDP in low and lower-middle income settings. [2, 3, 28, 31]. This should come as no surprise, as the majority of illnesses and complications that affect children under-5 are those that can be largely avoided by routine access to pediatric and post-natal services. Malnutrition and infectious diseases like malaria are particularly lethal for young children, with both of these issues increasing in severity as socioeconomic well-being declines. Further research may benefit from further breaking down under-5 mortality rates into subset rates such as infant mortality and neonatal mortality to even more clearly define areas of intervention. Countermeasures can help to reduce these impacts through food supplementation, growth monitoring, and comprehensive primary health care. Hopefully these estimates of the magnitude of the non-COVID-19 related child mortality can help marshal the resources needed to mitigate the burden.

## Supporting information

**S1 Appendix. Descriptive statistics.**
(ZIP)

**S2 Appendix. Multiple imputations.**
(ZIP)

**S3 Appendix. Estimation of the U5MR time trajectories at country level.**
(ZIP)

**S4 Appendix. Estimations—Model 1.**
(ZIP)

**S5 Appendix. Estimations—Model 2.**
(ZIP)

**S6 Appendix. Estimated Under-5 lives lost from 2020 downturns scaled from 5% to 15%.**
(ZIP)

**S7 Appendix. Sensitivity analysis of incremental deaths (95% confidence intervals).**
(ZIP)

## Acknowledgments

The COVID-Busters—a research team, formed by Dr. David Bishai—has provided invaluable feedback and support for this project. We also thank colleagues and students at the Johns Hopkins Bloomberg School of Public Health for their feedback and insight on this research.

## Author Contributions

**Conceptualization:** Marcelo Cardona, Joseph Millward, Alison Gemmill, Katelyn Jison Yoo, David M. Bishai.

**Data curation:** Marcelo Cardona.

**Formal analysis:** Marcelo Cardona, Joseph Millward, Alison Gemmill, David M. Bishai.

**Methodology:** Katelyn Jison Yoo.

**Writing – original draft:** Marcelo Cardona, Joseph Millward, Alison Gemmill, Katelyn Jison Yoo, David M. Bishai.

**Writing – review & editing:** Marcelo Cardona, Joseph Millward, Alison Gemmill, Katelyn Jison Yoo, David M. Bishai.

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
