## [Decision Letter · Decision Letter 0]

31 Aug 2021

PONE-D-21-22130

Estimated impact of the 2020 economic downturn on under-5 mortality for 129 countries

PLOS ONE

Dear Dr. Marcelo Cardona,

Thank you for submitting your manuscript to PLOS ONE. After careful consideration, we feel that it has merit but does not fully meet PLOS ONE’s publication criteria as it currently stands. Therefore, we invite you to submit a revised version of the manuscript that addresses the points raised during the review process.

We look forward to receiving your revised manuscript.

Kind regards,

Wen-Wei Sung, M.D., Ph.D.

Academic Editor

PLOS ONE

3. Please include a caption for figure 2.

Reviewers' comments:

Reviewer's Responses to Questions

**Comments to the Author**

1. Is the manuscript technically sound, and do the data support the conclusions?

Reviewer #1: No

Reviewer #2: Yes

2. Has the statistical analysis been performed appropriately and rigorously? 

Reviewer #1: No

Reviewer #2: Yes

3. Have the authors made all data underlying the findings in their manuscript fully available?

Reviewer #1: No

Reviewer #2: Yes

4. Is the manuscript presented in an intelligible fashion and written in standard English?

Reviewer #1: Yes

Reviewer #2: Yes

5. Review Comments to the Author

Reviewer #1: General Comments

1. The model is not well thought out.

2. I am not sure the model is correct.

3. The generalized linear mixed models are not well understood.

4. I wish I could have seen a portion of the data.

5. I have questions about the models. In particular � in equations 1 and 2.

SPECIFIC COMMENTS

Abstract: Each country’s individual slope relationship between child mortality and GDP was used to estimate the impact on U5MR of reductions in GDP per capita of 5%, 10%, and 15%.

RESPONSE: I encourage the authors to think about this sentence.

A 5% reduction in GDP per capita in 2020 was estimated to cause an additional 282,996 deaths in children under 5 in 2020. Recessions at 10% and 15% lead to higher losses of under-5 lives, increasing to 585,802 and 911,026 additional deaths, respectively. Nearly half of all the potential under-5 lives lost in LMICs were estimated to occur in Sub-Saharan Africa. Because most of these deaths will likely be due to nutrition and environmental factors amenable to intervention, countries should ensure continued investments in food supplementation, growth monitoring, and comprehensive primary health care to mitigate protentional burdens

RESPONSE: This section needs rewriting. Think about the results based on the models fit.

Introduction: The introduction can me more informative. For example, what is the Preston curve?

Methods Overview and Data Sources The methodology is presented in three sections. In section one, we present the methods used to re estimate and update Preston curves specific to each LMIC using data from 1990 to 2020. This provides multivariate adjusted estimates of the slope parameter relating GDP and Under-5 mortality individualized to each country’s most recent data along with 95% confidence intervals.

RESPONSE: This is not enough to help he reader understand what is being conveyed.

Multilevel Mixed Effects Multivariable Regression Analysis Regression analysis was used to estimate the Preston curve relationship between national income and under-5 mortality. First, we regressed the U5MR on GDP per capita and a set of socio- 6 economic covariates. A model-based approach using an iterative process was used to fill in missing values in the set of covariates. To estimate country-specific effects of a recession, we applied a multilevel mixed-effect linear regression to the relationship between GDP per capita and U5MR for each country. To control for heterogeneity across countries, the multilevel mixed-effect linear regression included fixed effects for a country’s region and income level. (Sensitivity analyses showed that results were not sensitive to inclusion of fixed effects.) A generalized log-linear model was estimated to ease the retransformation of impacts from a log-scale to natural units. Recession estimates were bracketed at 5%, 10%, and 15% reductions in country GDP per capita. Our baseline model to estimate the relationship between GDP per capita and U5MR had the following form: (1) 5 = + 1 + 

RESPONSE: Is this multilevel mixed model or generalized linear mixed models? I am a bit confused. Mixed model are for normal errors while generalized linear mixed models are for non-normal errors.

Lives Lost Estimation Country-specific estimates of 1 were then applied to GDP per capita data to predict an estimated mortality impact under the four different scenarios – no reduction in GDP per capita (scenario 1), 5% reduction (scenario 2), a 10% reduction (scenario 3), and 15% (scenario 4). We estimate potential recession-attributable loss of life by subtracting the deaths observed in scenario 1 from the projected number of deaths under scenarios 2-4

RESPONSE: I do not understand what this is supposed to convey.

Reviewer #2: Please refer to the attached document for my specific comments.

6. PLOS authors have the option to publish the peer review history of their article (what does this mean?). If published, this will include your full peer review and any attached files.

Reviewer #1: No

Reviewer #2: No

---

## [Author Response · Author response to Decision Letter 0]

3 Dec 2021

Response to reviewers

Editor comments

 We have edited the manuscript to ensure that it meets PLOS ONE’s style requirements.

 We have corrected this information.

 No changes are needed to the data availability statement.

4. Please include a caption for figure 2.

 We have added a caption for Figure 2.

Reviewer 1

1. The model is not well thought out.

The model we use is a standard way to empirically estimate the relationship between GDP and the under-5 mortality rate. Please see Preston SH. The Changing Relation between Mortality and Level of Economic Development. Popul Stud. 1975;29(2):231-248. doi:10.2307/2173509

2. I am not sure the model is correct.

We can assure to the reviewer that these models are widely used in the literature. See, for example, the following examples: 

• Debelew GT, Afework MF, Yalew AW. Determinants and causes of neonatal mortality in Jimma zone, southwest Ethiopia: a multilevel analysis of prospective follow up study. PloS one. 2014;9(9):e107184.

• Nelson DB, Moniz MH, Davis MM. Population-level factors associated with maternal mortality in the United States, 1997–2012. BMC public health. 2018;18(1):1–7.

• Fenta, S. M., & Fenta, H. M. (2020). Risk factors of child mortality in Ethiopia: application of multilevel two-part model. PLoS One, 15(8), e0237640.) 

We also note that the second reviewer did not have an issue with our empirical approach.

3. The generalized linear mixed models are not well understood.

As described above, these models are used to account for the hierarchical nature of the data (e.g., countries nested within regions.) We have added a citation to the paper that describes application of generalized linear mixed models.

4. I wish I could have seen a portion of the data.

Please see an extract of our dataset pasted below. The columns are as follows: country, country code, year, log U5MR, log GDP per capita, number of physicians per 1000 inhabitants, electricity consumption (Kilowatts per capita), the share of females in the national parliament, fertility rate and DPT immunization.

 

5. I have questions about the models. In particular � in equations 1 and 2.

We have added in the following description for equations 1 and 2:

1. “αj represents both the fixed and random intercept for country j.” – pg. 6

2. “ϵjt represents the error term for country j at time t.” – pg. 6

We hope that these additions help the reviewer understand our model.

SPECIFIC COMMENTS

Abstract: Each country’s individual slope relationship between child mortality and GDP was used to estimate the impact on U5MR of reductions in GDP per capita of 5%, 10%, and 15%.

RESPONSE: I encourage the authors to think about this sentence.

We have edited this to state the following: “Country-specific regression coefficients on the relationship between child mortality and GDP were used to estimate the impact on U5MR of reductions in GDP per capita of 5%, 10%, and 15%.” 

A 5% reduction in GDP per capita in 2020 was estimated to cause an additional 282,996 deaths in children under 5 in 2020. Recessions at 10% and 15% lead to higher losses of under-5 lives, increasing to 585,802 and 911,026 additional deaths, respectively. Nearly half of all the potential under-5 lives lost in LMICs were estimated to occur in Sub-Saharan Africa. Because most of these deaths will likely be due to nutrition and environmental factors amenable to intervention, countries should ensure continued investments in food supplementation, growth monitoring, and comprehensive primary health care to mitigate protentional burdens

RESPONSE: This section needs rewriting. Think about the results based on the models fit.

We appreciate the reviewer’s comment but believe that our interpretation of the results is faithful to the model-based scenarios we estimate.

Introduction: The introduction can me more informative. For example, what is the Preston curve?

We apologize for this oversight and can understand why our approach might have been confusing without first defining a Preston Curve, which is a core concept in the fields of Economics and Demography. We have added a paragraph to the introduction explaining the log relationship between GDP per capita and life expectancy first discovered by Samuel Preston, and have further cited studies that have shown correlation of GDP per capita with the under-5 mortality rate (U5MR). We also state in the Methods that whereas the original Preston uses life expectancy as a summary measure of health, in our study we use the under-5 mortality rate.

Methods Overview and Data Sources The methodology is presented in three sections. In section one, we present the methods used to re estimate and update Preston curves specific to each LMIC using data from 1990 to 2020. This provides multivariate adjusted estimates of the slope parameter relating GDP and Under-5 mortality individualized to each country’s most recent data along with 95% confidence intervals.

RESPONSE: This is not enough to help he reader understand what is being conveyed.

We have edited the methods section for clarity.

Multilevel Mixed Effects Multivariable Regression Analysis Regression analysis was used to estimate the Preston curve relationship between national income and under-5 mortality. First, we regressed the U5MR on GDP per capita and a set of socio- 6 economic covariates. A model-based approach using an iterative process was used to fill in missing values in the set of covariates. To estimate country-specific effects of a recession, we applied a multilevel mixed-effect linear regression to the relationship between GDP per capita and U5MR for each country. To control for heterogeneity across countries, the multilevel mixed-effect linear regression included fixed effects for a country’s region and income level. (Sensitivity analyses showed that results were not sensitive to inclusion of fixed effects.) A generalized log-linear model was estimated to ease the retransformation of impacts from a log-scale to natural units. Recession estimates were bracketed at 5%, 10%, and 15% reductions in country GDP per capita. Our baseline model to estimate the relationship between GDP per capita and U5MR had the following form: (1) 5 = + 1 + 

RESPONSE: Is this multilevel mixed model or generalized linear mixed models? I am a bit confused. Mixed model are for normal errors while generalized linear mixed models are for non-normal errors.

We appreciate the reviewer’s comment and we have address this comment on the first paragraph in sub-section 1.2. Moreover, the estimates are calculated using linear mixed-effects models, in which the overall error distribution of the model is assumed to be Gaussian (normal), and heteroskedasticity and correlations within lowest-level groups also may be modeled.

Lives Lost Estimation Country-specific estimates of 1 were then applied to GDP per capita data to predict an estimated mortality impact under the four different scenarios – no reduction in GDP per capita (scenario 1), 5% reduction (scenario 2), a 10% reduction (scenario 3), and 15% (scenario 4). We estimate potential recession-attributable loss of life by subtracting the deaths observed in scenario 1 from the projected number of deaths under scenarios 2-4

RESPONSE: I do not understand what this is supposed to convey.

𝑊e have reworded the latter part of this sentence on the bottom of pg. 7. In essence, we are conveying how excess mortality attributable to each potential economic downturn scenario (i.e. 5%, 10%, 15%) was derived for each country. We specifically state: “These estimates were then compared to scenario 1, which represents baseline under-5 mortality. By separately subtracting estimated deaths from scenarios 2 to 4 from those in scenario 1 (i.e. no GDP per capita reduction) we are able to provide estimates of additional lives lost that are attributable to each level of economic downturn.”

Reviewer 2

1. “Economic downturns have occurred in almost all countries…” Are we here already distinguishing between downturn and recession? If so, please define and distinguish between the two definitions at the outset.

Thank you for noting this. All use of the term “recession” has been replaced with the terminology “downturn” or “economic downturn” to better reflect terminology used in the paper title and introduction.

2. “The mechanisms relating distal social health determinants….” “Distal" seems like an odd choice of words here. Can this be replaced with something more descriptive?

We have removed the term distal from this sentence on pg. 2.

3. Please define the abbreviation [of GDP] at first use in the main text.

We now define GDP at first mention in the paper’s introduction section on pg. 2

4. “Parents securing supplemental income during economic hardship can subject children to less parental supervision heightening the risk of injury.” Please cite this assertion specifically.

We have added two citations for this statement.

5. “The pediatric community, therefore, plays an essential role in mitigating the health harms of sudden economic downturns…” Awkward phrasing here...suggest revising to something like "caregivers for children".

This sentence has been removed from the paper.

6. “…no study has estimated the indirect health effects of the 2020 economic downturn…” Please define here how you are distinguishing direct from indirect health effects.

We have provided explanation of the indirect nature of the effect of economic downturn on children’s health, and have further provided a citation for this point.

7. “…with an estimated 42–66 million additional children falling into extreme poverty.” Please define what is meant by "additional". Is this in addition to what was expected?

We have added clarity to this sentence by pulling additional description from the report cited by UNICEF to explain that the additional children falling into poverty in 2020 were compared to the estimated number of children living in extreme poverty during 2019 (pg. 3: lines 51-54). Specifically, the text reads: “The economic downturns of 2020 have also been projected to reverse a sustained trend of decline in global poverty, with an estimated 42–66 million additional children falling into extreme poverty on top of the estimated 386 million children in extreme poverty in 2019.”

8. “…for recessions as small as 5% and as large as 15%...” Does this refer to the proportional constriction of the economy?

We have clarified terminology to state that we are modeling reductions in GDP, defined as reductions in GDP in 2020, as compared to baseline GDP values in 2019. – pg. 3

9. “Recent studies have confirmed the adverse effect of recessions on under-5 mortality, showing that the impact in LMICs is three times larger than in countries with better economic indicators.” This last sentence seems like it should either be earlier in the introduction section or elsewhere (e.g., discussion section).

This sentence has been moved to the paragraph describing the Preston curve on pg. 2.

10. “…Preston curves specific to each LMIC…” This is the first mention of the "Preston curves". Suggest providing more information up front.

We apologize for this oversight. We now include more background information on Preston curves in the Introduction. – pg. 2

11. These last two sentences belong in the discussion section: “Admittedly, this method will offer a lower bound estimate of the full health impact of recession because the model excludes delayed mortality effects after one year, economically mediated deaths in adults, and non-fatal effects on health, social development, and cognition that are known to follow famines and adverse childhood experiences. However, estimates of just the mortality effects of the 2020 downturns can help alert policymakers of the need to plan additional efforts to mitigate the economic threats faced by vulnerable groups.” 

Thank you for this recommendation. We have moved this to the discussion section on pg. 7

12. “Because recessions in high income countries are known to have only negligible if any effects on child mortality…” This statement requires a reference. Also suggest softening the language so as not to suggest the complete absence of effects in highincome countries...just minimal ones (relatively speaking).

Citations have been added appropriately. Language modification to soften this statement has also been well-received and implemented.

13. “…2020 World Bank income classification requiring an income below 12,375 US$.” Please be precise about how income is defined here? For example, is this annual family income?

We have clarified that this classification is based on gross national income (GNI) per capita.

14. “A generalized log-linear model was estimated to ease the retransformation” What does it mean to "ease the retransformation"?

We have clarified what we mean by adding the following footnote: 

This specification has been used to have a linear relationship between U5MR and GDP per capita, and to represent the elasticity of U5MR with respect to GDP per capita. 

15. “We carried out additional analyses to examine the integrity and range of our estimates…” I'm not sure how this addresses the integrity of the estimates, which would seem to be driven more by the data sources and analysis rather than the range.

We have rephrased this opening paragraph to specify that we conducted sensitivity analysis to examine the uncertainty range of our estimates. – pg.5

16. “Table 2 presents results for the ten countries with the highest additional under-5 lives lost in 2020 under the four different scenarios.” Certainly all lives lost are meaningful, but would it not be relevant to first present as a proportion of the general population or of the under-5 general population?

Thank you for this recommendation. We have added a sentence to complement the analysis.

---

## [Decision Letter · Decision Letter 1]

26 Dec 2021

PONE-D-21-22130R1Estimated impact of the 2020 economic downturn on under-5 mortality for 129 countriesPLOS ONE

Dear Dr. Marcelo Cardona,

Thank you for submitting your manuscript to PLOS ONE. After careful consideration, we feel that it has merit but does not fully meet PLOS ONE’s publication criteria as it currently stands. Therefore, we invite you to submit a revised version of the manuscript that addresses the points raised during the review process.

We look forward to receiving your revised manuscript.

Kind regards,

Wen-Wei Sung, M.D., Ph.D.

Academic Editor

PLOS ONE

Journal Requirements:

Reviewers' comments:

Reviewer's Responses to Questions

**Comments to the Author**

1. If the authors have adequately addressed your comments raised in a previous round of review and you feel that this manuscript is now acceptable for publication, you may indicate that here to bypass the “Comments to the Author” section, enter your conflict of interest statement in the “Confidential to Editor” section, and submit your "Accept" recommendation.

Reviewer #2: All comments have been addressed

Reviewer #3: All comments have been addressed

2. Is the manuscript technically sound, and do the data support the conclusions?

Reviewer #2: Yes

Reviewer #3: Yes

3. Has the statistical analysis been performed appropriately and rigorously? 

Reviewer #2: Yes

Reviewer #3: Yes

4. Have the authors made all data underlying the findings in their manuscript fully available?

Reviewer #2: Yes

Reviewer #3: Yes

5. Is the manuscript presented in an intelligible fashion and written in standard English?

Reviewer #2: Yes

Reviewer #3: Yes

6. Review Comments to the Author

Reviewer #2: The authors provided thoughtful responses to the issues raised and revised the manuscript accordingly.

Reviewer #3: Generally, the article is well revised but it will require editing. The following issues should be revised;

1. Recessions of under-five mortality lead to… This should be written as recession of under-five mortality led to…

2. PROTENTIAL BURDEN should be written as potential burden

3. Estimates of uncertainty- authors should delete the empty brackets ().

4. LMIC countries. This should be LMIC but not LMIC countries.

7. PLOS authors have the option to publish the peer review history of their article (what does this mean?). If published, this will include your full peer review and any attached files.

Reviewer #2: No

Reviewer #3: No

---

## [Author Response · Author response to Decision Letter 1]

7 Jan 2022

Dear Reviewer #3,

We have addressed your comments as follows:

1. Recessions of under-five mortality lead to… This should be written as recession of under-five mortality led to…

 We have not been able to find the text “Recessions of under-five mortality lead to…” in the manuscript

2. PROTENTIAL BURDEN should be written as potential burden

We apologize for this typo, and we have edited it.

3. Estimates of uncertainty- authors should delete the empty brackets ().

We have not been able to find the empty brackets () in the Estimates of uncertainty sub-section of the manuscript.

4. LMIC countries. This should be LMIC but not LMIC countries.

We apologize for this oversight, and we have edited it.

Best regards,

Marcelo Cardona

---

## [Editor Report · Decision Letter 2]

17 Jan 2022

Estimated impact of the 2020 economic downturn on under-5 mortality for 129 countries

PONE-D-21-22130R2

Dear Dr. Marcelo Cardona,

We’re pleased to inform you that your manuscript has been judged scientifically suitable for publication and will be formally accepted for publication once it meets all outstanding technical requirements.

Kind regards,

Wen-Wei Sung, M.D., Ph.D.

Academic Editor

PLOS ONE
---

## [Editor Report · Acceptance letter]

28 Jan 2022

PONE-D-21-22130R2 

Estimated impact of the 2020 economic downturn on under-5 mortality for 129 countries 

Dear Dr. Cardona:

I'm pleased to inform you that your manuscript has been deemed suitable for publication in PLOS ONE. Congratulations! Your manuscript is now with our production department. 

Kind regards, 

on behalf of

Dr. Wen-Wei Sung 

Academic Editor

PLOS ONE